# ASSETFORMER: MODULAR 3D ASSETS GENERATION WITH AUTOREGRESSIVE TRANSFORMER

**Lingting Zhu**[1]  **Shengju Qian**[2*]  **Haidi Fan**[2]  **Jiayu Dong**[2]  **Zhenchao Jin**[1]
**Siwei Zhou**[2]  **Gen Dong**[2]  **Xin Wang**[2]  **Lequan Yu**[1†]
[1]The University of Hong Kong      [2]LIGHTSPEED
ltzhu99@connect.hku.hk, lqyu@hku.hk

## ABSTRACT

The digital industry demands high-quality, diverse modular 3D assets, especially for user-generated content (UGC). In this work, we introduce AssetFormer, an autoregressive Transformer-based model designed to generate modular 3D assets from textual descriptions. Our pilot study leverages real-world modular assets collected from online platforms. AssetFormer tackles the challenge of creating assets composed of primitives that adhere to constrained design parameters for various applications. By innovatively adapting module sequencing and decoding techniques inspired by language models, our approach enhances asset generation quality through autoregressive modeling. Initial results indicate the effectiveness of AssetFormer in streamlining asset creation for professional development and UGC scenarios. This work presents a flexible framework extendable to various types of modular 3D assets, contributing to the broader field of 3D content generation. The code is available at https://github.com/Advocate99/AssetFormer.

## 1 INTRODUCTION

3D asset generation has garnered significant attention due to its potential impact on digital creativity across various domains. Recent advancements have explored a variety of representations, including voxels (Brock et al., 2016; Wu et al., 2016), point clouds (Luo and Hu, 2021; Vahdat et al., 2022), neural fields (Gao et al., 2022; Chen and Zhang, 2019), and meshes (Siddiqui et al., 2024; Nash et al., 2020). However, despite progress in sophisticated geometry and texture, these traditional representations face critical limitations in real-world applications, particularly within

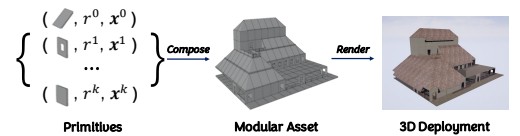

Figure 1: **Illustration of modular 3D assets.** Modular assets can be decomposed into primitives, each possessing its own attributes, e.g., the orientation $r$ and the position $x$. The modular asset can be rendered with configurations to enable 3D deployment.

the game industry. In professional game development, existing methods often struggle to meet the high-quality standards demanded by modern games, resulting in a time-intensive workflow for artists who may spend hundreds of hours meticulously designing and refining each asset. Meanwhile, in user-generated content (UGC) scenarios (Epic, 2017; Duan et al., 2022) and online gaming, these representations frequently yield large file sizes, which present substantial challenges for storage and transmission in efficiency-driven environments. Such issues can strain server infrastructure and hinder seamless sharing and real-time interaction—crucial elements in UGC platforms and multiplayer online games. Furthermore, the inherent complexity of these representations often restricts non-professional users from easily creating, modifying, and sharing their content, thereby limiting the potential for diverse and engaging user-generated game assets.

In digital production, artists frequently employ modules and constrained design spaces as foundational elements for complex assets. This approach, drawing concepts from Constructive Solid Geome-

---

*Project Lead.
†Corresponding author.

try (CSG) (Voelcker and Requicha, 1977; Laidlaw et al., 1986) in Computer-Aided Design (CAD), offers several advantages. It facilitates rapid prototyping, ensures asset consistency, and enables seamless integration into game engines. Utilizing CSG principles, artists can efficiently combine and manipulate basic shapes to create intricate forms with precision. This modular methodology not only streamlines the asset creation process but also lowers the barrier to entry for non-professional users, fostering broader participation (Krumm et al., 2008; Rymaszewski, 2007) and enhanced scalability in content creation. Moreover, this approach enables transmission efficiency in user-generated content (UGC) and online gaming environments.

While other 3D modalities benefit from the availability of growing public datasets (Deitke et al., 2023; Wu et al., 2023), modular 3D assets suffer from a significant scarcity of publicly available training data, leaving automatic modular asset generation an understudied field. This deficiency stems from the proprietary nature of most modular asset libraries, which are often closely guarded by game studios and content creators. To address this challenge, our research leverages modules and data collected from an online user generated content (UGC) platform, where players create intricate 3D homestead assets by manually arranging pre-defined construction materials.

Illustrated in Fig. 1, modular representation of 3D assets exemplifies the potential for complex asset creation from basic components but also highlights the demand for tools that can automate and enhance creation. Building on these insights, our study aims to develop a model capable of generating diverse modular 3D assets with customization on textual descriptions.

In this work, we propose a novel framework that leverages autoregressive modeling with modular 3D assets. Composed of primitive elements, each asset can be viewed as a series of modules, as well as proper decisions about their placement and orientation. This sequential nature aligns perfectly with autoregressive models, which excel at capturing and generating ordered sequences. Meanwhile, it mirrors the step-by-step process of human construction, leading to more intuitive and controllable asset generation. Unlike text or image generation, where the sequence order is often inherent (left-to-right for text, pixel-by-pixel for images), 3D assets pose a unique challenge in determining the optimal order of modular components. This ordering is crucial as it affects both the coherence of the generated structure and the model's ability to capture complex spatial relationships. By carefully analyzing the connectivity among primitives, we design improved tokenization algorithms and decoding strategies that capture the hierarchical and spatial relationships within assets. In summary, our contributions are as follows:

- We propose an autoregressive generation framework for modular 3D asset generation, which shows promising results compared to other 3D modalities.
- We introduce a large-scale dataset of modular 3D assets, collected and cleaned from the UGC platform of an online game. To our knowledge, this is the only real-world modular 3D dataset of high quality.
- We analyze the impact of module tokenization order and decoding strategies on the quality and diversity of generated assets, offering insights that can be extended to other 3D sequential generation tasks.
- Our model demonstrates the ability to generate high-quality, contextually appropriate 3D assets, providing a practical guide for the application of 3D generation.

## 2 RELATED WORK

**Generative Visual Modeling.** The recent years have seen a continuous pursuit of advanced generative models, including generative adversarial networks (GANs), autoregressive models (ARs), flows, and variational autoencoders (VAEs), and their crown battle for visual creation in image, video, and 3D applications (Goodfellow et al., 2014; Ho et al., 2020; Van Den Oord et al., 2016; Vaswani, 2017; Rombach et al., 2022; Chang et al., 2022; 2023; Kingma and Dhariwal, 2018; Kingma, 2013; Singer et al., 2022; Hong et al., 2022; Poole et al., 2022; Hong et al., 2023). Inspired by the scalability demonstrated by autoregressive models in language modeling (Achiam et al., 2023; Brown, 2020; Chowdhery et al., 2023), recent efforts have focused on extending the capabilities of AR models to mixed-modal modeling or challenging the dominance of diffusion models in visual generation (Zhu et al., 2023; Liu et al., 2024b; Team, 2024; Driess et al., 2023; Liu et al., 2024a; Wang et al., 2024a; Sun et al., 2024). For instance, Emu3 (Wang et al., 2024a) posits that next-token prediction is all you

need for achieving state-of-the-art performances in multimodal tasks, demonstrating robust results in the understanding and generation of images, text, and videos. Built upon autoregressive transformers, our work delves deeper into design rationales tailored to downstream visual creation for 3D assets, e.g., modular 3D generation.

**3D Generation.** Recent advancements in 3D generation have demonstrated significant progress, creating complex 3D representations from textual descriptions or sparse images (Gao et al., 2022; Lin et al., 2023; Poole et al., 2022; Tang et al., 2023; Zhang et al., 2024; Liu et al., 2023b; Long et al., 2024). These methods have explored various 3D representations, including voxels, point clouds, neural fields, and meshes. Notably, autoregressive Transformer-based models for mesh generation (Nash et al., 2020; Siddiqui et al., 2024; Chen et al., 2024c;b; Tang et al., 2024) have garnered attention due to their potential to synthesize detailed 3D structures. Despite these breakthroughs, existing methods face several challenges in real-world applications, including meeting high-quality standards, managing large file sizes, and providing accessibility for non-professional users. Some studies have adapted generative models for specific applications like CAD models (Wu et al., 2021; Li et al., 2022; Xu et al., 2024b; Ritchie et al., 2023) and human garments (Korosteleva and Lee, 2022; Liu et al., 2023a; He et al., 2024), aiming to address domain-specific challenges. Our work builds upon these advancements while specifically targeting the challenges of modular 3D asset generation, aiming to address the limitations of existing methods in terms of quality, efficiency, and accessibility.

**Autoregressive Modeling.** Autoregressive transformers have demonstrated remarkable success in language modeling and visual generation (Achiam et al., 2023; Liu et al., 2024b; Team, 2024; Liu et al., 2024a), benefiting from their scalability and ability to capture complex dependencies. However, adapting these models to visual and 3D domains presents unique challenges, particularly in data tokenization and sequence ordering. For instance, VQGAN (Esser et al., 2021) employs a codebook for images, while MAR (Li et al., 2024b) learns a continuous-valued space using diffusion-based probability distribution modeling. In the 3D domain, methods for mesh generation (Tang et al., 2024; Chen et al., 2024d) have explored compact mesh tokenization to effectively represent complex 3D structures. LLaMA-Mesh Wang et al. (2024b) and Mesh-LLM Fang et al. (2025) integrate LLM's strong prior knowledge and enable the generation of text-serialized 3D meshes, but struggle producing very complex 3D meshes. A recent work (Ye et al., 2025) uses AR model for decomposing complex shapes into 3D primitive, with part-level understanding (Mo et al., 2019; Gao et al., 2021; Hertz et al., 2022; Li et al., 2024a). Furthermore, decoding strategies (Holtzman et al., 2019; Leviathan et al., 2023; Chen et al., 2023a; Teng et al., 2024) for improving generation quality and inference speed remain an active area of research in visual models.

## 3 METHOD

### 3.1 PROBLEM FORMULATION

We collect the intricate 3D assets from an online UGC platform, which the users manually create with the provided modular materials. As the data represents distinctive homesteads, each asset comprises a sequence of building primitives, e.g., roof and floor patches, where the building primitive has its attributes including class $c \in \mathcal{C}$, rotation $r \in \mathcal{R}$ (vertical axis), and position $x \in \mathcal{X}^3$, with their finite sets of discrete values. To be specific, $i$-th sample can be characterized as $N_i$ primitives $\{P_j\}_{j=1}^{N_i}$ and $P_j = (c_j, r_j, x_j)$. Our goal is to learn a generative model $G$ capable of synthesizing samples from textual description $t$: $G : t \to \{P\}_{i=1}^{N}$.

The dataset source is obtained and cleaned from real user-created assets, which are of high complexity and variety. One advantage of the modular 3D representation is its easy compatibility with traditional Procedural Content Generation (PCG) methods. To better study the influence of different data sources, we formed another data source with PCG in addition to the real data we collected. Building on procedural generation (Short and Adams, 2017; Raistrick et al., 2023), we use random generators for attributes such as the number of storeys and the positions of key modules. The details of the Algorithm can be found in Appendix. To prepare the text prompt, we use GPT-4o (OpenAI, 2024) to produce phrase bundles such as *(apartment, multi-story, flat roof, few windows)*, characterizing the global features of the asset based on the rendered images.

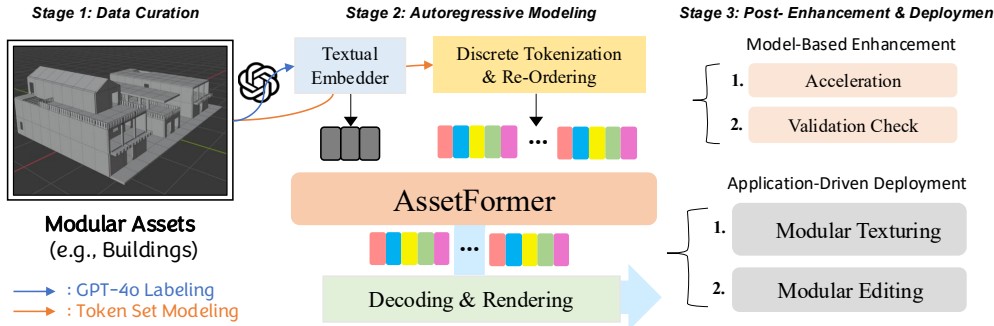

Figure 2: **Overview of the AssetFormer Framework.** Given the modular assets, e.g., the building, we first render the assets in digital engines and produce the images for querying GPT-4o. The cleaned captions, pre-filled with a re-ordered token set, serve as input for the autoregressive modeling. After training, AssetFormer autoregressively produces modular assets that are ready to be integrated into industrial environments, with model-based enhancement and application-driven deployment.

## 3.2 AUTOREGRESSIVE TRANSFORMER MODELING

To model the sequence distribution of tokens, our AssetFormer is built on a Decoder-only Transformer, using standard cross-entropy loss for next-token prediction:

$$\mathcal{L} = \text{CrossEntropy}(\text{Shift}(\hat{S}), \text{Tokenize}(\{P\})), \tag{1}$$

where $\text{Shift}(\hat{S})$ denotes shifted result of predicted tokens sequence $\hat{S}$ and $\{P\}$ represents the asset comprising primitives. We adopt Llama (Touvron et al., 2023) as the Transformer backbone with our vocabulary and model configurations, and use 1D rotary positional embeddings (Su et al., 2024). The text features are projected to tokens and pre-filled to the token sequence during training and inference.

**Discrete Tokenization.** Modular 3D assets typically consist of primitives with discrete attributes and fixed decision spaces. This inherent discreteness allows us to leverage a more efficient representation without resorting to complex graph encoders like those used in MeshGPT (Siddiqui et al., 2024). Our approach utilizes finite sets of discrete values, directly modeling pre-defined vocabularies for each attribute type in a lossless manner. Each asset is represented as a sequence of token tuples, where the $i$-th sample has a primitive length of $N_i$ and a token length of $5N_i$, reflecting the five parameters required for each attribute tuple. Following common practice (Team, 2024; Liu et al., 2024a; Wu et al., 2021; He et al., 2024), these sequences are padded with <EOS> tokens to indicate the end of the prediction.

**Token Set Modeling.** Each primitive is defined by 5 parameters and jointly modeled with a transformer, necessitating the maintenance of distinct vocabularies for different attributes. The combined set of attribute vocabularies, along with the <EOS> token, forms the token vocabulary $\mathcal{V}$:

$$\mathcal{V} = \mathcal{C} \vee \mathcal{R} \vee \mathcal{X}_0 \vee \mathcal{X}_1 \vee \mathcal{X}_2 \vee \{< \text{EOS} >\},$$
$$|\mathcal{V}| = |\mathcal{C}| + |\mathcal{R}| + |\mathcal{X}_0| + |\mathcal{X}_1| + |\mathcal{X}_2| + 1, \tag{2}$$

where $\mathcal{C}, \mathcal{R}, \mathcal{X}_0, \mathcal{X}_1, \mathcal{X}_2$ denote the vocabulary of primitive class, rotation, and 3D positions, respectively. Consequently, the raw token sequence $T$ is expressed as:

$$T = \{c^0, r^0, x_0^0, x_1^0, x_2^0, \dots, c^{n-1}, r^{n-1}, x_0^{n-1}, x_1^{n-1}, x_2^{n-1}, \text{EOS}\}, \tag{3}$$

where $n$ denotes the number of primitives. While this joint vocabulary approach does not affect training, as we can naively treat the periodic token sequences conventionally for next-token prediction, it requires special consideration during inference. To achieve diversity and randomness, we sample from the logits distribution for each token. However, this can potentially produce tokens that do not belong to the current attribute's vocabulary. For instance, after generating a token $c \in \mathcal{C}$ (primitive class), the next token should be drawn from $\mathcal{R}$ (rotation). To ensure valid token set decoding, we filter out unwanted logits and re-normalize the remaining non-zero distribution.

**Token Re-Ordering.** Token order plays a crucial role, as emphasized by recent studies in Transformer-based visual models (Yu et al., 2024; Tang et al., 2024; Chang et al., 2022; Chen et al., 2024d). 3D assets contain rich structural information both globally and locally. To capture this hierarchical and spatial relationship, we design traversal methods based on depth-first search (DFS) and breadth-first search (BFS). These methods ensure modular connectivity locally while maintaining a first-to-end sequential order globally. In industrial practice, the graph traversal is often used in validation check where the key nodes are checked with designed connectivity rules. This scenario is eligible for checking the validity of the generated building in post-processing, where the popped nodes of the stack in DFS are required to follow certain rules with the neighborhood nodes.

In practice, we start at the lower corner of the asset and traverse all primitives using a graph searching method. This produce a permutation order $\mathcal{A} = \{\tau_0, \tau_1, ..., \tau_{n-1}\}$ for a primitive set of length $n$, where $\tau_i$ denotes the original index of the $i$-th element in the raw primitive sequence. Consequently, the re-ordered token sequence $T'$ is given by:

$$T' = \text{ReOrder}(T) = \{c^{\tau_0}, r^{\tau_0}, x_0^{\tau_0}, x_1^{\tau_0}, x_2^{\tau_0}, \ldots, c^{\tau_{n-1}}, r^{\tau_{n-1}}, x_0^{\tau_{n-1}}, x_1^{\tau_{n-1}}, x_2^{\tau_{n-1}}, \text{EOS}\}. \quad (4)$$

While both DFS and BFS can capture local features with modular connectivity, it is not immediately clear which method leads to better data normalization. Empirically, we have found that DFS performs slightly better as the primitive re-ordering method. This re-ordering facilitates the training of the token set modeling, and remains agnostic to asset deployment in rendering.

**Classifier-Free Guidance.** Inspired by the widely used Classifier-Free Guidance (CFG) (Ho and Salimans, 2022) in text-to-image diffusion models (Saharia et al., 2022; Xue et al., 2024; Chen et al., 2023b), which enhances generation fidelity and text alignment, recent research on generative visual Transformers has also adopted it to achieve similar goals. We follow the methodology outlined in (Liu et al., 2024a; Sun et al., 2024), randomly dropping control signals in training and utilizing unconditional logits additionally during inference. The decoding process is based on logits calculation: $l_{cfg} = l' + s \cdot (l - l')$, where $l$ and $l'$ denote the conditional and unconditional logits, and $s$ denotes the CFG scale.

### 3.3 Autoregressive Transformer Decoding

As large language models advance, generative visual Transformers can significantly benefit from shared techniques adapted for visual tasks. We aim to present a preliminary analysis of sampling techniques that affect AssetFormer's quality. Furthermore, since our modular representation allows for seamless integration into game engines or rendering pipelines, without the necessity for post-processing steps like vertex merging as required in MeshGPT (Siddiqui et al., 2024), we implement decoding techniques to significantly enhance on-the-fly asset generation.

**Sampling Strategies.** AssetFormer generates modules sequentially to form complete assets, starting with pre-filled text tokens and continuing until the <EOS> token is generated. While we've explored various sampling strategies including greedy search, beam search, and top-k sampling (Fan et al., 2018), we find that top-k sampling offers a good balance between asset quality and diversity.

**SlowFast Decoding.** To address the computational challenges of autoregressive decoding, we introduce SlowFast decoding, our adaptation of speculative decoding (Leviathan et al., 2023; Chen et al., 2023a) for 3D asset generation which accelerates decoding without compromising quality and requires minimal additional training. Our SlowFast decoding employs two models:

- The draft model with smaller capacity to quickly predict easy tokens.
- The target model with larger capacity to handle more complex token predictions.

The effectiveness of SlowFast decoding in modular 3D asset generation stems from the varying complexity of different parts of the asset. Many modular locations, especially those following common patterns or simple structures, can be accurately predicted by the smaller, faster model. The larger, slower model is then used to decode more challenging tokens that require a deeper understanding of context or complex spatial relationships. This approach is particularly suited to our modular representation, as it allows for efficient prediction of common or simple components while ensuring accurate generation of more intricate or context-dependent parts of the asset. Our implementation includes modifications to filter out unwanted logits of other token types (Nash et al.,

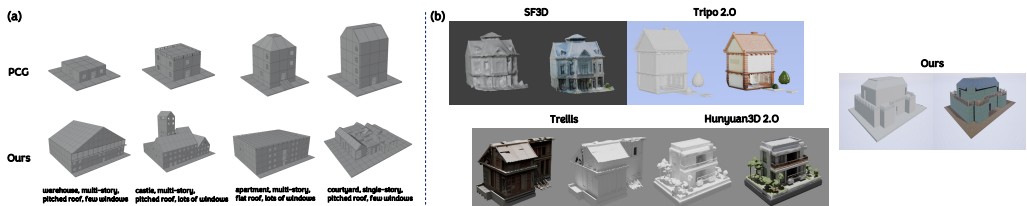

Figure 3: **Qualitative comparison with comparison methods.** (a) While PCG can synthesize high-quality building models, it requires meticulous algorithm design for complex buildings and can only produce simple assets that are difficult to control with text. (b) Compared with 3D generation methods, which typically yield dense meshes, struggle to accurately capture intricate geometries (the internal structure of buildings), and produce imperfect textures, our methods follow the design rationales of preferred rules (e.g., with standard primitives of plain faces) and deliver precise texture in real-world pipelines with primitive-texture mapping.

2020) during reject sampling, similar to our token set modeling approach. The detailed SlowFast decoding algorithm is presented in Algorithm 2 in the Appendix.

# 4 EXPERIMENTS

## 4.1 DATASET

Our dataset is derived from two sources: procedurally-synthesized data using PCG (detailed in Algorithm 1 in the Appendix) and real user-created 3D assets collected from the game. We streamlined the user data by removing extraneous long-tail information and mapping the assets to a set of 25 basic primitives. To ensure dataset quality, we employed a combination of automatic GPT-4o (OpenAI, 2024) queries and manual review to filter out overly simple and duplicate samples. This process resulted in a high-quality dataset comprising 16,000 real samples and 4,000 synthesized samples. The average token length of the data sample is larger than 4,000. For DFS and BFS, we select the random node as the next query if multiple nodes available which is equivalent to randomly sorting the data sample and query the first node in multiple choices. Note that our focus is on learning the modular arrangement of 3D assets, with texture considerations typically left for post-processing during production. We employ a total of 25 primitives, which can be broadly categorized into three types: roof primitives, wall primitives, and other component primitives.

To enable text control over asset generation, we utilize GPT-4o to generate phrase bundles that indicate the global type of assets and highlight key features. It's worth noting rendering data exhibits a significant domain gap compared to natural images, making it challenging to caption discriminative global types for buildings using multimodal language models. Nevertheless, the generated captions provide probabilistic guidance for our model. Detailed information about the modular primitives, prompt templates, and phrase statistics can be found in the Appendix.

## 4.2 IMPLEMENTATION DETAILS

The joint vocabulary serves as the discrete token space for our Transformer model, with a total vocabulary size $|\mathcal{V}|$ of 214. This comprises $|\mathcal{C}| = 25$ (primitive classes), $|\mathcal{R}| = 4$ (rotations), $|\mathcal{X}_0| = 59$, $|\mathcal{X}_1| = 44$, and $|\mathcal{X}_2| = 81$ (3D positions). Complex data samples contain up to 1,000 primitives each. To enable text control, we follow (Sun et al., 2024) to use FLAN-T5 XL (Chung et al., 2024) as the encoder and project the features through an MLP (Chen et al., 2023b). To support CFG, we implement a condition dropout ratio of 0.1 during training. Our primary model, AssetFormer-B (312M), uses a Llama-based backbone (no pre-trained weights) consisting of 24 Transformer layers. To facilitate SlowFast decoding, we additionally train a smaller draft model, AssetFormer-S (87M) with 12 Transformer layers. For inference, we employ a CFG scale of 2.0 and a temperature of 0.7, using top-k sampling with k=10 for all comparisons.

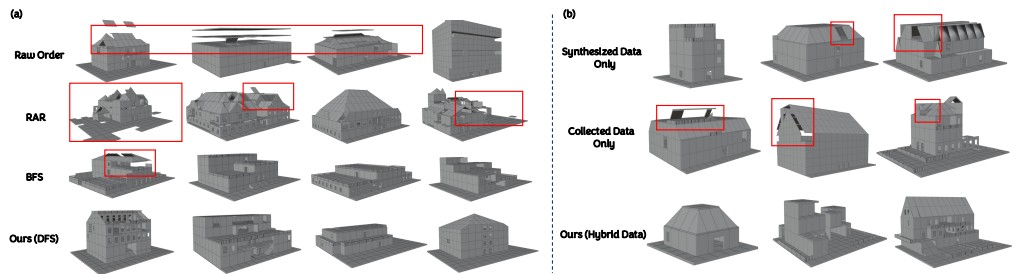

Figure 4: **Qualitative ablation analysis.** (a) Ablation on token orders. With improper token order, the model struggles to fit and generate the distribution accurately. (b) Ablation on data sources. The models fail to cover a wide range of diverse building types and exhibits a higher ratio of failure cases when trained on a single data source. The artifacts are indicated in red rectangles.

## 4.3 COMPARISON WITH THE BASELINES

PCG techniques have long been the cornerstone of game production pipelines, creating assets using PCG. While these methods are well-established, they lack more free-form control with challenges from generative methods. In this section, we include a procedural generation method as a baseline, using the same algorithm employed in our data synthesis stage. This baseline randomizes modular features such as orientation and position but lacks complex modeling and textual control. Fig. 3(a) demonstrates AssetFormer's ability to generate a variety of assets in a data-driven manner, controlled by text conditions, which is not present in the PCG method.

We also compare our method with state-of-the-art general 3D generation approaches, specifically SF3D (Boss et al., 2024), Tripo 2.0 (Tripo, 2024), Trellis (Xiang et al., 2024), and Hunyuan3D 2.0 (Zhao et al., 2025) which are designed to generate dense meshes for open-domain objects. We use the prompt *"A high-quality 3D model of a building"* [1] or their official image generation integration to generate images for image-to-3D pipelines. Fig. 3(b) highlights the visual results of these methods alongside our approach. By adopting a primitive-based representation, Asset avoids the generation of low-quality, dense meshes that are difficult to integrate into industry pipelines. Although recent 3D generation methods have achieved significant improvements in producing high-quality geometry, they continue to exhibit noticeable texturing artifacts. These issues primarily stem from the suboptimal performance of current texturing techniques (Zhao et al., 2025; Zhu et al., 2025; Youwang et al., 2024). In contrast, primitive-based generation methods benefit from the more advanced development of primitive-texture mapping, which results in more refined texturing outcomes.

Table 1: **Quantitative results compared with baselines.** We show comparison results on generation quality, indicated by FID and CLIP score.

| Methods | FID ↓ | CLIP ↑ |
|---|---|---|
| True Data | / | **0.322** |
| PCG (Algorithm 1) | 108.476 | 0.319 |
| AssetFormer + Greedy Search | 63.351 | 0.319 |
| AssetFormer + Beam Search | 63.333 | **0.321** |
| AssetFormer + Top-K Sampling | **55.186** | 0.320 |

To perform a quantitative evaluation, we assess generation quality using Fréchet Inception Distance (FID) (Heusel et al., 2017; Parmar et al., 2022) and CLIP score (Radford et al., 2021). Our evaluation procedure involves synthesizing 500 assets using sampled test prompts and rendering 500 images from a fixed viewpoint that properly captures the global structure. FID is computed between these rendered images and the full training set, with *clean-FID* (Parmar et al., 2022), which indicates the visual quality of generated assets. Due to the difficulties of large-scale rendering, the FID values are much higher than those typically seen in text-to-image works (Liu et al., 2024a; Chang et al., 2023) as

---

[1]https://blackforestlabs.ai

they are computed with the smaller sets, but the relative values faithfully indicate similarity and thus the quality of the generated buildings since the numbers are large enough and controlled the same. CLIP score is computed between rendered image features and the text feature of a fixed prompt, i.e., *"A high-quality 3D model of a building"*. We opt not to use CLIP scores between images and generation prompts due to challenges in obtaining informative results (originated from the domain gap) in our unusual image and text domains, i.e., all settings produce fluctuating results near 0.29, yet these relative performances poorly align with human validation.

Table 1 presents the quantitative results. While the PCG method can synthesize compact modular assets, it struggles to cover the full breadth of the data distribution and generate richly detailed outputs with sophisticated structures. This is reflected in the FID scores, given that our training data includes both simple and complex assets. Regarding sampling strategies, our quantitative results indicate that top-k sampling outperforms both greedy search and beam search.

We further compare our method with MeshGPT (Siddiqui et al., 2024), which utilizes mesh representation and leverages a Transformer as the decoder. Please check the detailed analysis and discussion in the Appendix.

### 4.4 Ablation Studies

#### 4.4.1 Ablation study on Token Orders

The ablation study on token orders demonstrates the effectiveness of our proposed primitive token re-ordering method. Table 2 presents a comparison of different ordering operations. The results clearly indicate that re-ordering methods, specifically DFS and BFS, yield superior results compared to learning sequences in their raw order.

Table 2: **Quantitative ablation analysis on token orders.** We compare the results of models trained on different token orders and we also implement a recent token randomized training method design for autoregressive modeling of image generation.

| Ordering Techniques | FID $\downarrow$ | CLIP $\uparrow$ |
| --- | --- | --- |
| Raw Order | 65.215 | 0.318 |
| RAR (Yu et al., 2024) | 83.561 | 0.313 |
| Breadth-First-Search | 61.620 | 0.319 |
| Depth-First-Search | **55.186** | **0.320** |

We also implement RAR (Yu et al., 2024), a recent work focusing on token randomization in training text-to-image autoregressive models. RAR employs an annealing strategy and tailored positional embedding to outperform standard raster-order-based AR image generator training. To adapt RAR to our setting, we use a hierarchical operation to re-order tokens in an annealing manner, accommodating our token set modeling for building primitives. Specifically, given primitives in DFS order, we randomly permute them while freezing the second-stage permutation, maintaining the original order of attribute tokens within each primitive. Interestingly, our results indicate that RAR does not perform well in our task. We hypothesize that, unlike images which benefit from token disturbance for better bidirectional learning, the challenges in leveraging local details of 3D structures hinder the efficient learning capabilities.

Note that while CLIP scores may be similar in absolute values, the visual results of the baselines can be poor, as illustrated in Fig. 4(a). Clear artifacts, highlighted in red rectangles, can be observed. A notable phenomenon is the presence of isolated generated parts in results obtained with raw order, reinforcing our conclusion that re-ordering helps grasp local structures and ensure modular connectivity.

#### 4.4.2 Ablation Study on Data Sources

Our method incorporates data from both procedural generation and human creation. Table 3 presents metrics for models trained with different data sources, revealing an intriguing phenomenon. Models trained solely on collected data show substantial improvement over those using only synthesized data,

with FID scores of 63.381 and 113.560, respectively. However, the most striking result emerges from the combination of both data sources, yielding a superior FID of 55.186. This improvement likely stems from the complementary nature of the two data types. Synthesized assets, generated through PCG, tend to be more compact and structured. While they may perform poorly in isolation due to limited diversity, they provide a beneficial scaffolding for the model's learning process. In contrast, user-created data offers greater diversity and randomness, which, when combined with the structured synthesized data, enhances the model's ability to generalize.

Table 3: **Ablation analysis on data sources.** We train models on different configurations of data sources, and show the distribution difference of data generation.

| Training Data Types | FID ↓ | CLIP ↑ |
|---|---|---|
| Synthesized Data Only | 113.560 | 0.320 |
| Collected Data Only | 63.381 | **0.321** |
| Synthesized Data + Collected Data | **55.186** | 0.320 |

Fig. 4(b) illustrates this synergy, showcasing rendered results from various generated assets. The visualization highlights how the integration of both data sources enables the model to capture a wider spectrum of architectural styles and structures, overcoming the limitations observed when relying on a single data type. Our findings underscore the importance of leveraging multiple, complementary data sources in training generative models for modular assets. The structured nature of synthesized data provides a solid foundation, while the diversity of collected data expands the model's creative range. This balanced approach not only improves the quality and variety of generated buildings but also enhances the model's robustness in meeting diverse user preferences.

### 4.4.3 ANALYSIS ON SLOWFAST DECODING

We implement SlowFast decoding for autoregressive asset generation, which required training an additional draft model. This draft model, with its smaller capacity and reduced number of parameters, enables accelerated decoding through meticulously designed algorithms (Leviathan et al., 2023; Chen et al., 2023a). Table 4 presents the generation quality and decoding speed for models with varying parameters, controlled by the number of Transformer layers, heads, and feature dimensions. AssetFormer-B is the base model we have trained and the smaller AssetFormer-S is the draft model. These results demonstrate that our tailored SlowFast decoding method successfully accelerates the generation process without sacrificing performance.

The SlowFast decoding is particularly effective for modular 3D asset generation, where prediction difficulty varies significantly. Simple primitives and standard components are swiftly handled by the draft model, while complex, context-dependent elements benefit from the larger model's nuanced predictions.

Table 4: **Analysis on SlowFast Decoding.** We train models of different parameters and perform SlowFast decoding. The generation quality and decoding speed are evaluated.

| Model Configurations | FID ↓ | Speed (token/s) ↑ |
|---|---|---|
| AssetFormer-S (87M) | 60.420 | 151.31 |
| AssetFormer-B (312M) | 55.186 | 80.62 |
| SlowFast Decoding | 55.831 | 119.02 |

### 4.4.4 ANALYSIS ON MODULAR REPRESENTATION VERSUS NATIVE 3D REPRESENTATIONS

In our work, a key design and exploration lies in the modular representation, which is well-suited for autoregressive modeling and UGC deployment. Recent text-to-3D methods have demonstrated promising native 3D generation capabilities, leveraging representations such as VecSet Zhang et al. (2023); Zhao et al. (2025); Zhang et al. (2024); Chen et al. (2024a; 2025) or sparse voxel grids Li et al. (2025); Wu et al. (2025); Xiang et al. (2024); He et al. (2025). Nevertheless, the state of the arts still still face challenges in generating high-quality structural details—particularly for internal structures—primarily stemming from limited training data and suboptimal data preprocessing pipelines. A critical bottleneck is that most existing 3D generative methods require a watertight

Figure 5: **Qualitative analysis on fine-tuning native 3D generative models.** (a) After Watertight conversion, the modular information is lost and the geometry erroneous (e.g., the ladder). (b) The geometry details are actually changed (zoom in to see the vertices and faces). (c) The fine-tuned Hunyuan3D 2.1 produces an overall inferior assets and (d) the details are poor.

geometry step, which introduces additional complexity to geometry processing and prevents VAEs from recovering high-fidelity details. While native 3D generation demands more advanced data curation strategies to enable finer-grained control, the modular representation we propose serves as a practical alternative for real-world applications. Further details and comparisons are provided in Appendix A.2 and Table 6.

In this section, we further demonstrate that the modular representation facilitates high-fidelity asset production: it preserves geometric details and yields visually consistent, high-quality results. To highlight the limitations of existing text-to-3D approaches in downstream tasks, we design a controlled toy experiment. Specifically, we use our dataset—structured as modular representations—to export object geometries (vertices and faces), then apply watertight preprocessing to generate training data for native 3D generative models. Visual comparisons of assets before and after the watertight step are presented in Fig. 5(a). It is observed that object conversion and watertight processing lead to loss of modular information, as individual primitives are merged into a single unstructured mesh. Additionally, this process alters fine-grained details (see Fig. 5(b)), distorting the geometry of primitives sourced from our asset library—where structural integrity is intentionally preserved. Furthermore, we argue that the existing native 3D models (e.g., Hunyuan3D 2.1) struggle to generate high-quality outputs when trained on our modular data—particularly for complex objects with internal structures. To validate this, we conduct an *overfitting* experiment: we fine-tune Hunyuan3D 2.1 on a small subset of 10 modular samples, then evaluate its performance using the exact training conditions for inference. Fig. 5(c) shows the output meshes from the fine-tuned model, while Fig. 5(d) highlights the poor reconstruction of training sample details. These results confirm that even with overfitting, the base model fails to capture the complex structure of modular data: it corrupts individual primitives, ultimately limiting the practical utility of native 3D generation for modular-based applications.

## 5 CONCLUSION

In this work, we introduce **AssetFormer**, a novel autoregressive Transformer-based framework designed for modular 3D asset generation. Our approach emphasizes the modeling of assets from primitives and the learning of their distribution for generative applications. The framework is meticulously tailored to accommodate both potential applications and user-generated content (UGC), ensuring versatility and adaptability in various contexts. We innovatively adapt token sequencing and decoding techniques inspired by language models, achieving high-fidelity asset generation through autoregressive modeling. We anticipate that AssetFormer will contribute significantly to the evolving landscape of 3D content creation and enable widespread real-world applications.

**Limitations.** Currently, AssetFormer is designed to accept only text input for asset generation. The ability to incorporate image-based conditioning remains uncertain and unexplored. Meanwhile, our model relies on fixed discrete vocabularies, necessitating additional design considerations to accommodate varing design spaces.

ACKNOWLEDGEMENTS

This work was supported in part by the Research Grants Council of Hong Kong (C5055-24G, and T45-401/22-N), and the Hong Kong Innovation and Technology Fund (GHP/318/22GD).

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

# A APPENDIX

## A.1 USER STUDY

We conducted a user study to better validate the qualitative performance of our method. The study involved 6 participants aged between 22 and 28 years. The participants were asked to grade the buildings based on four criteria: *compactness*, *diversity*, *aesthetic*, and *complexity*. The grading was done in batches, each consisting of six building samples. We included ground truth data, PCG generated data, and the synthetic data from AssetFormer. The results, shown in Table 5, are rated on a scale of 1 to 5, with 5 being the highest. Our participants widely acknowledged that our method produces high-fidelity results in terms of diversity, aesthetic, and complexity. It is worth noting that the PCG method, which generates buildings with simpler structures, received higher grades for compactness from participants, even surpassing the ground truth due to the different domains.

Table 5: **User study results.** The ratings of compactness, diversity, aesthetic, and complexity, are on a scale of 1-5.

| Method | Compactness | Diversity | Aesthetic | Complexity |
|---|---|---|---|---|
| Ground Truth | *3.83* | *4.00* | *3.67* | *4.42* |
| PCG | **4.47** | 2.42 | 3.33 | 2.08 |
| AssetFormer | 3.42 | **3.50** | **3.50** | **3.92** |

## A.2 COMPARISON WITH MESHGPT

We select MeshGPT (Siddiqui et al., 2024), which utilizes mesh representation and leverages a Transformer as the decoder, as a baseline for qualitative comparison. We further discuss the characteristics of both mesh representation and modular representation, shown in Table 6. Additionally, we attempted to directly fine-tune language models with supervised fine-tuning on our building JSON data. Given that the building data can comprise up to 1000 primitives, requiring more than 3K tokens when tokenized from the raw JSON data which includes the primitives types and attributes, we adopted LongLoRa (Chen et al., 2023c) to fine-tune Llama-2 (Touvron et al., 2023). However, the results are far poorer than AssetFormer, with a qualitative winrate below 5%. We account for the inherent complexity and implicit nature of the representation, which pose significant challenges for language-based models to comprehend and create long sequences.

We present the comparison results with MeshGPT in Fig. 6. Although we generate results in modules with AssetFormer, it is important to note that these results can be seamlessly converted to triangle meshes if needed, as the modules are compact, as shown in Fig. 6. For this comparison, we first converted all building data to triangle meshes and extracted the vertex and face information required by MeshGPT. We then trained the autoencoder and Transformer on our data using MeshGPT. We present both non-transparent and transparent rendered results. While MeshGPT encodes faces and vertices and learns mesh generation based on face and vertex representation, it becomes evident that as the task complexity increases, i.e., generating complex buildings with numerous vertices and faces, the training becomes challenging and the decoding often fails. We do not include subsequent works like MeshAnything (Chen et al., 2024c) and MeshXL (Chen et al., 2024b) as they adopt the same representation. Additionally, since modules can be decomposed into triangle meshes, modular representation is more efficient and requires fewer tokens compared to mesh-based generation methods. Even recent works focusing on compact tokenization of meshes, such as EdgeRunner (Tang et al., 2024), typically handle meshes with fewer than 4K faces, whereas our data can comprise more than 30K faces in triangle meshes.

Table 6: **Comparison of mesh representation and modular representation.**

| Representation | Lossless | Ready for Engines | Triangle Meshes | Efficient | No Post-Processing | User-Friendly |
|---|---|---|---|---|---|---|
| Mesh | ✔ | ✔ | ✔ | ✗ | ✗ | ✗ |
| Modular | ✔ | ✔ | ✔ | ✔ | ✔ | ✔ |

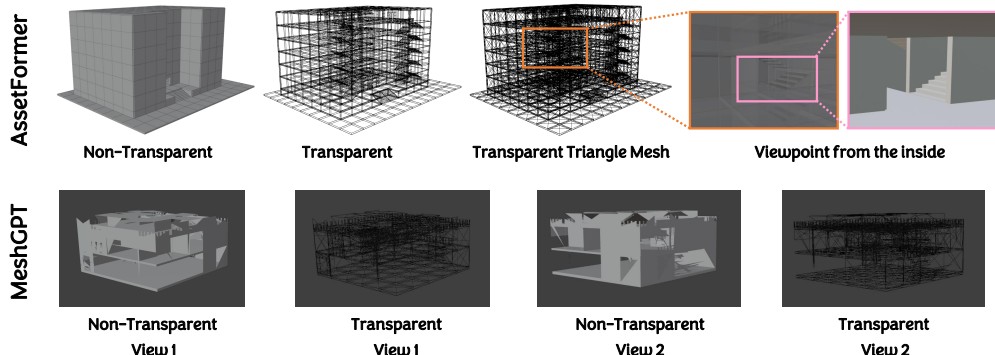

Figure 6: **Qualitative comparison with MeshGPT.** We present the non-transparent and transparent results of MeshGPT and ours. Our method can produce compact arrangement of primitives, as demonstrated by the transparent rendering images and the viewpoint from the inside.

Table 6 presents the characteristics of mesh-based and modular representations. Meshes benefit from not requiring representation conversion in 3D generation methods that adopt Triplane and implicit representations (Xu et al., 2024a; Poole et al., 2022), which has recently increased their popularity. It is worth noting that modular representation also inherits the crucial strengths of meshes. Built upon primitive modules, the representation is lossless, ready for game engines, and can be directly converted into triangle meshes.

Furthermore, Fig. 7 showcases the X-Ray results, revealing internal structures of buildings. The results demonstrate that AssetFormer is capable of synthesizing buildings with not only impressive appearances but also intricate internal structures. It is important to note that the internal structure of game assets is crucial for real-world applications. While AssetFormer excludes explicit texture information, the modular nature of our generated geometry allows for versatile streamlined applications. We showcase the versatility of modular representation in Fig. 8, by mapping the primitives to a diverse set of textured intricate modules. This flexibility aligns with industry practices and enables seamless integration with specific scenes or game aesthetics. Moreover, such modular representation supports both procedural and generative texture rendering techniques, allowing for dynamic and diverse visual outcomes.

Mesh representation can hardly compete on special needs in real-world scenarios. Using mesh representation to train generative models presents a key issue: the token length can be extremely long, especially for real-world objects with details. Additionally, after decoding, mesh representation requires post-processing to merge close points in 3D space, whereas modules are compactly connected. Furthermore, although mesh representation is ready for artists, it is not yet user-friendly. In contrast, modular representation-based generation serves as a powerful technology integrated for user-generated content (UGC), thanks to its user-friendly manipulation.

### A.3 ALGORITHM DESCRIPTION

**Procedural Content Generation**. PCG is effective for quickly synthesizing simple buildings and can produce data samples without artifacts. However, this method struggles to adapt to a variety of complex buildings, resulting in a mismatch with user preferences and a gap in representing intricate building distributions. We randomly set attributes such as width, length, and floor height. Using roof primitives, wall primitives, and component primitives, we can randomly generate walls, floors, and roofs, and decorate each aspect with special primitives, such as doors and stairs.

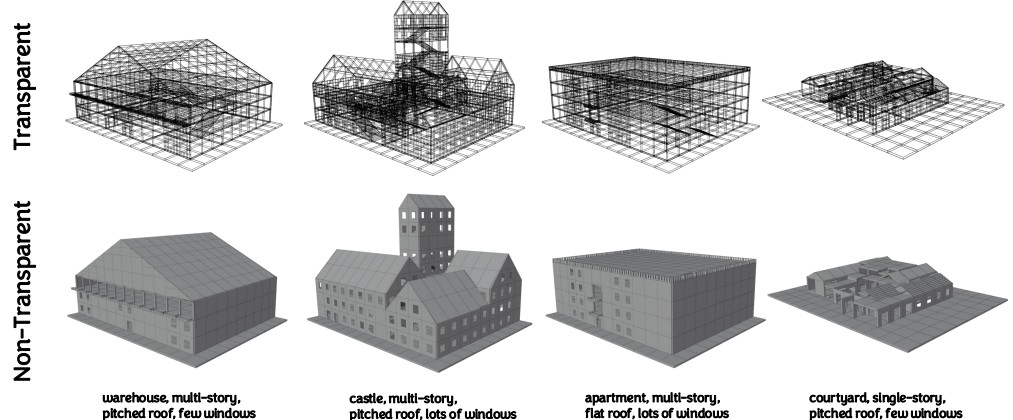

Figure 7: **X-Ray results of the generated buildings.** We further present transparent results that highlight the complex and compact structures of the generated samples. For supplementary illustration, we provide results from different viewpoints of the buildings in Fig. 3.

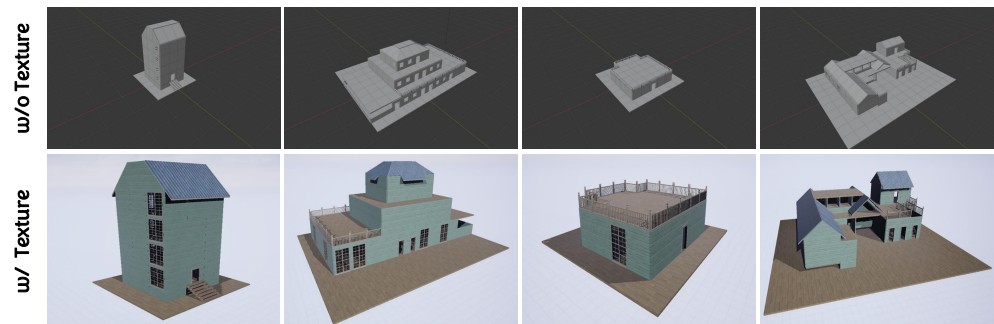

Figure 8: **Visualization of generated buildings with textured intricate modules mapping.** Our generated assets can be seamlessly integrated into engine runtime, mapped with different textured modules of Level-of-details. We show various viewpoints for reference.

---

**Algorithm 1** Procedural Content Generation

---

1: width = $\text{Randint}(1, MAX\_WIDTH)$
2: length = $\text{Randint}(1, MAX\_LENGTH)$
3: floorHeight = $\text{Randint}(1, MAX\_FLOOR\_HEIGHT)$
4: // Decorate with floor primitives and stair primitives
5: $\text{SetWall}(width, length, floorHeight)$
6: // Decorate with floor primitives and stair primitives
7: **for** $floor\_id$ in Range(MAX_FLOOR_HEIGHT) **do**
8:     $\text{SetPlane}(width, length, floor\_id)$
9: **end for**
10: // Decorate with roof primitives
11: $\text{SetRoof}(width, length, floorHeight)$

---

---

**Algorithm 2** SlowFast Decoding

---

1: **Require** the target model AssetFormer-B which produces $q(\cdot|\cdot)$, the draft model AssetFormer-S which produces $p(\cdot|\cdot)$
2: **Input** the text prompt, which gives pre-filled tokens $prefix$, the lookahead $K$, and target sequence length $T$
3: Set token number $n = 0$
4: **while** $n < T$ **do**
5:     // Sample from draft model
6:     **for** $t$ in $\text{Range}(K)$ **do**
7:         $\hat{x}_t \sim p(x|prefix, x_0, \ldots, x_{n-1}, \hat{x}_0, \ldots, \hat{x}_{t-1})$
8:     **end for**
9:     // Forward target model
10:     Compute logits $q(x|prefix, x_0, \ldots, x_{n-1}, \hat{x}_0, \ldots, \hat{x}_t), t = 0, \ldots, K - 1$
11:     // Drop with a probability of 1-q/p
12:     $reject\_pos = \text{RandomDrop}(p\_logits, q\_logits)$
13:     // Get the primitive types for the rejected tokens
14:     $reject\_type = \text{GetTokenType}(n, reject\_pos)$
15:     // Draw from q-p as Speculative Sampling with primitive token type awareness
16:     $resampled\_tokens = \text{Sample}(reject\_pos, reject\_type, q\_logits, p\_logits)$
17:     Sample $x_{n+K}$ from $q$ if needed
18:     Update $n$
19: **end while**
20: **Return** $[x_0, \cdots, x_{T-1}]$

---

**SlowFast Decoding.** The SlowFast Decoding method, is adapted from Speculative Decoding (Chen et al., 2023a; Leviathan et al., 2023), which utilizes two models of different sizes to accelerate the sampling of large language models. Following this key insight, we train a draft model, AssetFormer-B, to quickly produce draft tokens. After decoding with the draft model, the target model processes the token sequences to obtain logits, which are used to reject existing tokens with a defined probability. Notably, since our modular representation requires meaningful token orders, we also need to track the vocabulary types for each token. With the tracked types, we filter out the logits that do not belong to the current token and sample within the re-normalized distribution. Experiments have clearly demonstrated the effectiveness of SlowFast Decoding, achieving acceleration without compromising performance.

### A.4 EMERGENT EDITING CAPABILITIES

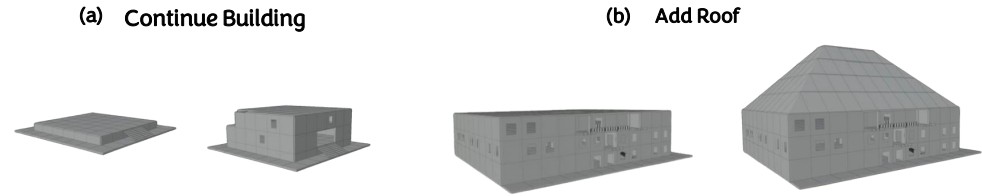

Figure 9: **Illustration of emergent editing of AssetFormer.** We showcase that without further training, the model is able to edit the modular buildings. The case (a) and case (b) show that the model can continue building and add roof. The two prompts are *"small building, single-story, flat roof, minimal windows"* and *"modern building, multi-story, pitched roof, lots of windows"*.

We further demonstrate that our model enables zero-shot editing of modular buildings, as illustrated in Fig. 9. Framed as a sequence inpainting task, this application showcases the model's ability to extend existing modular structures and incorporate roof components. By pre-training the model on text-to-modular building data, it learns both the structural constraints and semantic relationships inherent to modular architectures. In practice, given a modular building representation, we first perform DFS-based token reordering as preprocessing, using these reordered tokens as the initial sequence. Additionally, unwanted primitives (e.g., existing roof structures) can be removed, with the remaining tokens serving as the target for inpainting. Notably, despite not being explicitly trained for

this editing task—nor exposed to the distinct token order patterns of the inpainting setup—the model successfully predicts the missing target primitives and completes the editing task.

## A.5 DIVERSITY FOR THE SAME PROMPT

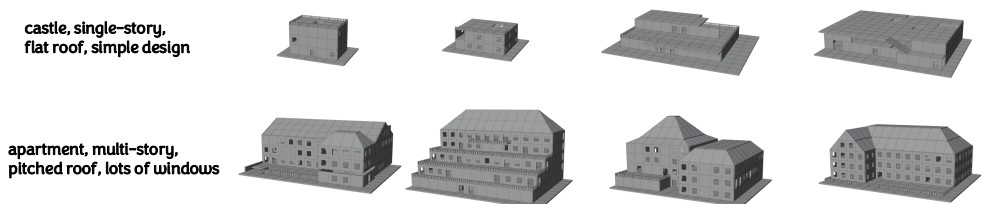

Figure 10: **The generated results with the same prompts.** We showcase that the generated assets with the same prompts to show the diversity. The two rows are the cases of two different prompts.

We present generated results with the same prompts in Fig. 10. The cases show that with the same prompts, benefitting from the sampling of the Transformer, the generated samples are diverse.

## A.6 MORE ASSETS: GALLERY IN UNREAL ENGINE

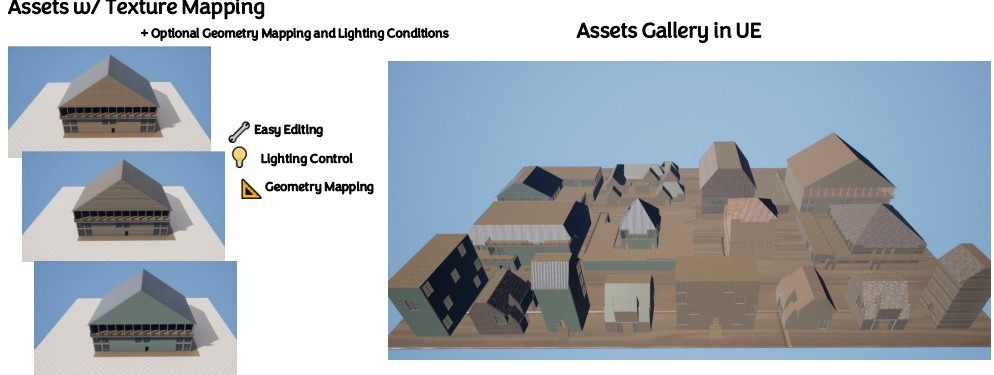

Figure 11: **Asset gallery in UE.** We showcase that the generated assets can be easily edited and seamlessly integrated into Unreal Engine (UE), enabling the assembly of cohesive and production-ready gallery collections.

We present additional generated results in Fig. 11. The modular representation natively enables texture mapping and even geometry mapping. Notably, the generated samples—equipped with customizable textures, optional geometry mapping, and adjustable lighting—can be seamlessly integrated into Unreal Engine, directly supporting real-world 3D content production workflows.

## A.7 MORE INFORMATION ON THE DATA

**Modular Primitives.** Fig. 12 illustrates the primitives utilized in our data. These categories are displayed in three separate columns. Additionally, we provide statistics of the primitives in Fig 13. We present the distributions of the PCG data and the collected real data to show the distribution differences in the dataset.

**Prompt Curation.** To prepare the text conditions for the building samples, we use GPT-4o (OpenAI, 2024) to generate text descriptions based on rendered images from a fixed viewpoint. To control the flexibility of the text conditions, we use curated prompts, as presented in Table 7. Additionally, we provide statistics of the text phrases in Fig 14. We present the distributions of the PCG data and the collected real data to show the distribution differences in the dataset.

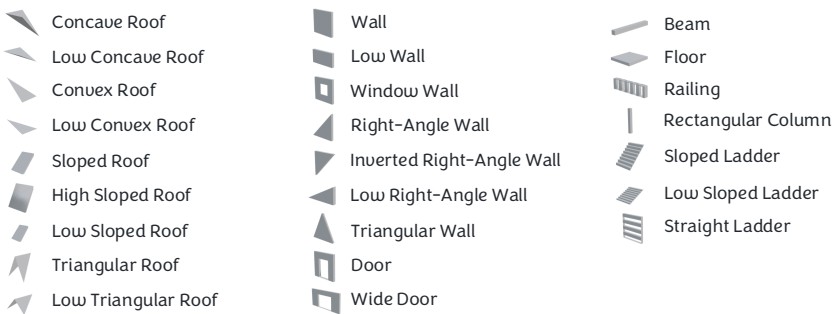

Figure 12: **Descriptions of primitives in building data.** We showcases roof primitives, wall primitives, and other component primitives in three columns.

Table 7: **The prompt of querying GPT-4o.**

I will provide an image of a building and you should generate one string, which comprises of several phrases.

The first phrase describes the building type:
Example phrases and rules:
'castle': often with features like towers or battlements.
'skyscraper': tall, rectangular building.
'courtyard': not high, often with an open space in the middle.
'mansion': large, impressive, featuring multiple stories and high-end architectural details.
'townhouse': usually multi-story, often in a row with similar houses.
'apartment': multi-story, often with lots of windows.
If all the above types are not precise, you can use other types.

The second phrase should describe the height of the building in terms of floors:
Example phrases: 'single-story', 'multi-story', 'high-rise'.
If the building is higher that 5 stories, it can be classified as 'high-rise'.

The remaining two phrases, giving the most precise features of the building. You do not need to always use some phrases if they are ambiguous. For example, if there are more than 15 windows, then you can use 'lots of windows' but do not treat doors as windows.
Example phrases: 'pitched roof', 'flat roof', 'lots of windows', 'magnificent', 'dull', 'weird', etc.

## A.8    VIDEO VISUALIZATION

We include a video demo of AssetFormer in our supplementary materials, including the generation process of the method, the comparison of token orders, and the integration of generated game assets into game engines.

To further compare the token order, which is a key factor in our methodology design, we present the video visualization demonstrating the construction sequence of results generated by different token orders. We construct the building primitives according to the decoding sequence and show the construction videos for raw orders, BFS, and DFS.

Additionally, we demonstrate the seamless integration of modular representation-based generated building data into game engines. To this end, we develop a lightweight software on Unreal Engine, a mature game engine in the gaming industry. By placing the modular building in a scene, we can control a digital human to traverse the scene, surrounding the building and entering it, as if the building is part of the normal environment in games. This is especially appealing when we map the building assets to have complex geometry and texture details with asset mapping. The video demo clearly demonstrates that our generated game assets are effective, convenient, clean, and ready for integration into game engines.

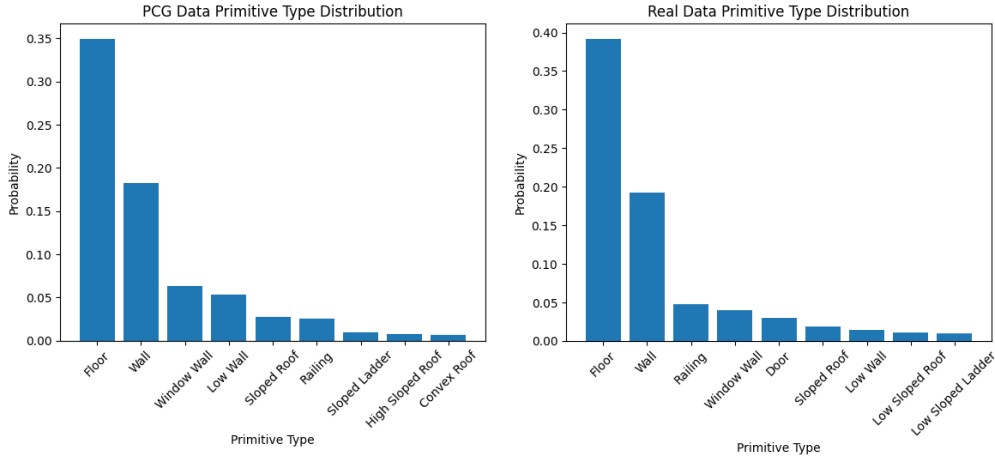

Figure 13: **Phrases statistics of primitives.** We show the histograms on primitives of PCG data and real data.

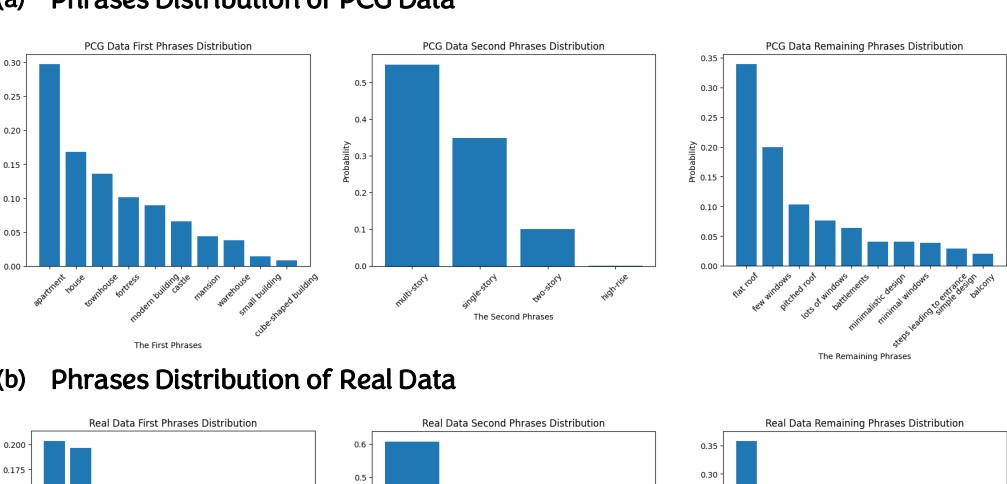

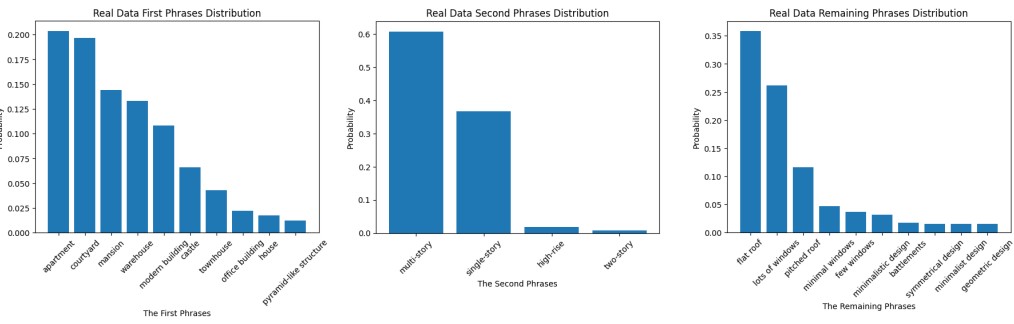

Figure 14: **Phrases statistics of text phrases.** We show two histogram sets in (a) and (b) on text phrases of PCG data and real data. The three histograms in one set show the distribution of the first phrases, the second phrases, and the remaining phrases, used in our dataset.

## A.9 LLM USAGE DISCLOSURE

We used GPT-4o exclusively for grammar checks and light editing. The LLM acted as a writing aid and was not involved in generating ideas or content, thus not qualifying as an author.

