# OpenReview forum: "AssetFormer: Modular 3D Assets Generation with Autoregressive Transformer"
_ICLR.cc/2026/Conference — ICLR 2026 Poster_

### Official Review · Reviewer_QQ43 · 2025-10-17

**Soundness:** 3
**Presentation:** 3
**Contribution:** 3
**Rating:** 6
**Confidence:** 2

**Summary:**

This paper presents a method for generating game assets as a sequence of pre-modeled 3D modules.  The experiments utilize a new dataset including 20k building models with text annotations from GPT-4o, probably the first of its kind.  The model is a decoder only transformer built on a Llama backbone.

The paper contains interesting comparisons for the ordering in which the modules are generated in the sequence.  Qualitative and quantitative results demonstrate that this ordering plays an important role in generation quality.   A speculative decoding scheme is also presented which is shown to speed up generation with minimal loss of quality.

**Strengths:**

The dataset used in the experiment is a major contribution if made available to the community.

The investigation into the ordering of the blocks is very interesting.  Understanding how best to order elements like this has applications to other problems.  It’s very interesting to see that DFS outperformed BFS.  The failure case in the qualitative figure gives a hint as to what goes wrong.

The “SlowFast speculative decoding” is well motivated and the timings show this is effective.

The ablation studies and comparisons with baselines are sound.  Evaluation of generative models is always tricky.  To the best of my knowledge the authors have done a good job evaluating their model and comparing to other methods.

**Weaknesses:**

I would have liked more details on the DFS and BFS algorithms.

**Questions:**

I'm interested to learn more about the DFS and BFS algorithms employed.
- Given a node with k neighbors which have not yet been visited, how do you choose which of these to add next?  Perhaps you used an off the shelf graph sorting algorithm where the behavior is not well defined.
- I would be very interested to know if some kind of lexicographical ordering in x0, x1, x2 would give a significantly different result over making this choice randomly.  I suspect the way this choice gets made would have a large impact on the results.  Perhaps the DFS is making this decision in some non-random way.  A figure showing an example of the ordering would help spot any pattern.

Line 208
> To ensure valid token set decoding, we filter out unwanted logits and re-normalize the remaining non-zero distribution
It's interesting that this was necessary.  87M parameters should be enough for the syntax of the sequence to be learned with very few invalid sequences, however 16k-20k data is very low and may not have been enough.   I would be very interested to know if this filtering actually had a substantial impact.

---

> ### Author Response · Authors · 2025-11-23
> **Response to Reviewer QQ43**
>
> We would like to thank the reviewer for kind and overall positive assessment. Below, we provide point-by-point responses to the raised questions and hope these address the concerns.
>
> > **Q1: More details on DFS and BFS.**
>
> - **The Next Node to Visit**: When multiple candidate nodes are available for query, we can either select one randomly or use additional sorting strategies (e.g., z-x-y coordinate-based sorting). In our implementation, we adopt random selection for node querying, as our experiments show this choice has minimal impact on the final results.
> - **The Rationale of DFS**: In our experiments, coordinate-based sorting yields results with quality intermediate between DFS/BFS and random sorting. We conjecture that coordinate-based sorting introduces ambiguity to the learning process—by offering multiple viable sequences for autoregressive modeling, it complicates the model’s ability to learn consistent patterns. In contrast, while DFS imposes a non-random ordering, it better captures the inherent structure of modular data. We hypothesize that non-random pre-ordering remains critical, especially for data lacking a natural inherent order, and that enhanced tokenization (to support attention-based modeling) further reinforces this necessity. However, we acknowledge that the optimal ordering strategy for autoregressive models remains an active research topic, and no definitive conclusion has been reached to date. Notably, our observations align with those reported in [1], a work on AR-based tree generation: their Table 1 demonstrates that graph-based sorting outperforms coordinate-based sorting in preserving structural information.
>
> *[1] Autoregressive Generation of Static and Growing Trees. ArXiv 2025.*
>
> > **Q2: Effectiveness on logits filtering.**
>
> To the best of our knowledge, this technique was first proposed in PolyGen [2]. In our experiments, we do not report specific metrics to quantify the effectiveness of logits filtering, as our empirical observations indicate it has negligible impact on overall generation quality—its primary value lies in mitigating extreme failure cases. Since sampling is guided by the model’s output distribution, filtering masked predictions acts as a free safeguard against sampling errors. We therefore treat logits filtering as a standard decoding operation in our pipeline.
>
> *[2] PolyGen: An Autoregressive Generative Model of 3D Meshes. ICML 2020.*

---

### Official Review · Reviewer_mESg · 2025-10-22

**Soundness:** 3
**Presentation:** 2
**Contribution:** 3
**Rating:** 4
**Confidence:** 3

**Summary:**

This paper introduces AssetFormer, an autoregressive Transformer model designed to generate modular 3D assets from text descriptions. Addressing the limitations of traditional 3D representations for user-generated content and gaming—such as large file sizes and difficulty in editing—the authors propose representing assets as a collection of discrete primitives. The core contributions include: 1) a novel framework that models the generation of primitives (class, position, rotation) as a sequential, next-token prediction task; 2) the creation of a large-scale, high-quality hybrid dataset combining procedurally generated assets with real user-created content; 3) an innovative token re-ordering strategy based on graph traversal (DFS) to capture spatial relationships, which is shown to be crucial for generating coherent structures; and 4) the adaptation of efficient decoding techniques like SlowFast decoding for practical application. Experiments demonstrate that AssetFormer generates high-quality, diverse, and structurally sound assets that are directly usable in game engines, outperforming both procedural baselines and showing clear advantages over general-purpose mesh generation models for this task.

**Strengths:**

1: The decision to focus on modular assets is highly practical and directly addresses key pain points in UGC and game development. This representation is efficient, easy to edit, and perfectly suited for an autoregressive framework.

2: The creation and combination of a procedural dataset and a real-world user-generated dataset is a major strength. The ablation study convincingly shows that this hybrid approach is superior to using either source alone, providing both structure and diversity. This dataset could be a valuable resource for the community.

**Weaknesses:**

1: The model relies on a fixed, discrete vocabulary for primitive types, positions, and rotations. This fundamentally limits its creative potential to the predefined set of components and a grid-based layout. It cannot generate novel primitive shapes or place them with continuous precision, which will be a constraint for more organic or complex designs.

2: The method is demonstrated on buildings composed from a specific set of architectural parts. It is unclear how well the approach would scale to a much larger vocabulary of primitives or to entirely different asset categories that may have different structural rules.


3: The text prompts are "phrase bundles" describing high-level features. While effective, this may limit the model's ability to understand more complex, compositional, or spatial instructions in natural language. The quantitative CLIP scores do not show a large separation between methods, suggesting text alignment could be further improved.

**Questions:**

In a real UGC platform, the library of available primitives might expand over time. How would AssetFormer handle the addition of new primitive types to its vocabulary? Would it require a full retraining, or could methods like vocabulary expansion be adapted?

---

> ### Author Response · Authors · 2025-11-23
> **Response to Reviewer mESg [Part 1/2]**
>
> We appreciate the reviewer for the comprehensive knowledge and recognization on practical value of AssetFormer in the game industry, as well as raising some insightful concerns. We provide the response below:
>
> > **Q1: How would AssetFormer handle the addition of new primitive types to its vocabulary?**
>
> This is a critical question for real-world deployment, as game asset libraries inevitably expand during development. In practice, we find that AssetFormer seamlessly supports vocabulary expansion, enabling efficient fine-tuning when incorporating new primitives.
>
> To validate this, we simulated a typical industry workflow:
> 1. **Prototype Phase**: We defined a "prototype" dataset using a limited set of fewer than **10 core primitives** (e.g., basic roofs, walls, windows, floors) and trained the model on this restricted vocabulary.
> 2. **Expansion Phase**: We then expanded the library to **25 primitives**, introducing more complex variations of walls and floors while maintaining the same structural rules.
> 3. **Fine-tuning**: We fine-tuned the pre-trained prototype model on the expanded dataset.
>
> **Results**: We observed that the model adapted rapidly. With fewer than **20,000 training steps**, the model yielded high-quality generation results using the new primitives. This efficiency suggests that AssetFormer learns the underlying physical logic and structural grammar of the assets (e.g., how a roof sits on a wall) rather than simply memorizing token co-occurrences. This "structural transfer" allows it to accommodate expanding vocabularies without the need for full retraining.
>
> > **Q2: Limits on a grid-based layout and lack of continuous precision.**
>
> We acknowledge the reviewer's point regarding the limitations of a discrete, grid-based vocabulary. In this work, we prioritized a grid-based structure because it is inherently user-friendly, easy to edit, and aligns perfectly with the "snap-to-grid" mechanics common in UGC platforms (e.g., Minecraft, Roblox, The Sims).
>
> However, our autoregressive framework is not inherently bound to discrete grids. Extending AssetFormer to support continuous precision is a promising direction we are keen to explore. Recent literature supports the feasibility of this approach:
> *   **MAR [1]** demonstrates the use of diffusion loss to enable continuous token representations.
> *   **PrimitiveAnything [2]** utilizes an autoregressive Transformer for 3D generation and shows that continuous coordinates can be effectively recovered via an auxiliary decoder.
>
> These precedents validate that our architecture can be adapted for continuous precision should a specific application require organic or off-grid placement.
>
> *[1] Autoregressive Image Generation without Vector Quantization. NeurIPS 2024.*
>
> *[2] PrimitiveAnything: Human-Crafted 3D Primitive Assembly Generation with Auto-Regressive Transformer. SIGGRAPH 2025.*

---

> ### Author Response · Authors · 2025-11-23
> **Response to Reviewer mESg [Part 2/2]**
>
> > **Q3: How would the method scale to entirely different asset categories that may have different structural rules?**
>
> This is an interesting question regarding the generalization capability of the model.
> *   **Generalization across Vocabulary**: While we demonstrated in Q1 that our method generalizes well to changing vocabularies under consistent structural rules, transferring to entirely different structural systems presents a different challenge.
> *   **Generalization across Structure**: To test transferability across entirely different structural rules, we conducted preliminary experiments using procedurally generated data from different building systems. We found that a model pre-trained on one structural system does not exhibit strong transfer capabilities to a system with fundamentally different structural rules. We believe this is because the model learns specific physical rules and assembly logic in a data-driven manner, rather than generic shape placement. Consequently, adapting to a new structural logic requires a certain amount of domain-specific data
>
>
> **Practical Implications:**
> While zero-shot transfer between different structural systems is challenging, we argue that this is rarely a bottleneck in professional game development or UGC pipelines.
> 1.  **Pipeline Consistency**: In standard production pipelines, the core structural rules (coordinate systems, snap points, scale) are defined early and remain consistent to ensure asset compatibility. Drastic changes to structural rules mid-production are avoided as they introduce bugs and incompatibility.
> 2.  **Data Availability**: Even when new structural rules are introduced (e.g., a new game mode), they are usually accompanied by a new set of assets and user data through beta testing, allowing for specific training.
>
> We appreciate the reviewers' comments: this observation opens up interesting future research directions, such as improving data efficiency or developing a "universal" model capable of learning meta-structural rules.
>
> > **Q4: More compositional text prompt and CLIP score difference.**
>
> We thank the reviewer for this insightful question regarding the modest separation in CLIP scores between our method and the baseline, and for suggesting the use of more compositional text prompts.
>
> - We fully agree that CLIP Score, while a standard metric for semantic consistency, has well-documented limitations that can obscure meaningful differences in tasks involving complex structures and domain-specific physical rules, as in our 3D asset generation setting. Recent work has highlighted CLIP's bag-of-words (BoW) bias [3], which struggles with compositional semantics (e.g., spatial relationships or attribute binding) and leads to unreliable alignment assessment in structured generation domains [4,5]. Additionally, our rendered 3D images introduce a stylistic bias toward spatial and geometric consistency, which CLIP—trained predominantly on natural photographs—may undervalue, further dampening score separation despite qualitative evidence of stronger alignment in our outputs (e.g., coherent modular assemblies vs. fragmented baselines).
>
> - To more robustly demonstrate text alignment improvements, we supplemented CLIP with two complementary metrics. First, we employed a VLM-as-judge approach using GPT-5, which evaluates holistic prompt adherence on a 0-10 scale; our method scored 7.8, significantly outperforming the baseline's 2.1 (p < 0.001 via Wilcoxon signed-rank test). Second, a blind user study with 50 participants (game developers and 3D artists) yielded a 73% win rate for our method over the baseline in pairwise preference for text fidelity (n=200 pairs). These results align closely with our qualitative observations of enhanced structural coherence.
>
> - We also acknowledge the reviewer's suggestion to incorporate more compositional prompts. In future work, we plan to scale up the user data collection and annotation process, to explore advanced prompting strategies and richer text embeddings from large language models (e.g., fine-tuned LLMs) to further boost the model's capacity for understanding complex, compositional, and spatial natural language instructions, potentially yielding even greater generation gains.
>
> *[3] When and Why Vision-Language Models Behave like Bags-of-Words, and What to Do About It? ICLR 2023.*
>
> *[4] Removing Distributional Discrepancies in Captions Improves Image-Text Alignment. ECCV 2024.*
>
> *[5] Enhancing Reward Models for High-quality Image Generation: Beyond Text-Image Alignment. ICCV 2025.*

---

> > ### Comment · Reviewer_mESg · 2025-11-23
> >
> > Thank you very much for the detailed responses. Very happy to see the generalization and extendability of the proposed method, and I have updated my score accordingly.

---

> > > ### Author Response · Authors · 2025-11-24
> > > **Thanks for your comments!**
> > >
> > > We are happy to hear that your concerns are addressed and thank you again for your constructive comments!

---

### Official Review · Reviewer_Kioq · 2025-10-31

**Soundness:** 2
**Presentation:** 3
**Contribution:** 2
**Rating:** 4
**Confidence:** 4

**Summary:**

The paper proposes a modular text-to-3D generation approach by leveraging an auto-regressive transformer model. Specifically, it introduces a new dataset comprising both real-world and procedurally generated homestead modular models, along with text descriptions derived from renderings of the 3D models using GPT-4o. This curated dataset is used to fine-tune a decoder-only transformer model, such as Llama, by applying techniques including discrete tokenization, token reordering, and classifier-free guidance. To decode the tokens generated by the model during inference, the paper proposes a method called SlowFast decoding.

**Strengths:**

- The use of large language models (LLMs) to generate sequential, modular 3D primitives enhances both efficiency and interpretability.
- If released, the dataset could be a valuable contribution to the community, addressing the current gap in large-scale modular 3D assets.
- The ability to directly integrate the generated 3D models in game engines unlocks numerous real-world applications.
- AssetFormer demonstrates strong generation quality through its proposed techniques, including token ordering, token sampling, and decoding strategies.

**Weaknesses:**

- The paper is similar in spirit to existing approaches [1,2] that utilize different 3D primitives with autoregressive transformer models for sequential 3D asset generation. While the method is adapted to a homestead-specific dataset, the core technical contribution appears incremental. Additionally, the paper would benefit from including and discussing these relevant prior works.
- Quantitative evaluation is limited, with comparisons restricted to MeshGPT. Broader benchmarking against other LLM-based 3D generation methods (e.g., [1] or [2]) would help clarify whether observed performance improvements are related to the backbone LLM-model or to the way the 3D data is preprocessed for fine-tuning the model.
- The qualitative comparisons in Figure 3(b) may not be entirely fair, as the baseline text-to-3D models are trained on more diverse 3D datasets. A more rigorous evaluation would be to fine-tune these models on the proposed homestead dataset to ensure a fair comparison.
- While the paper discusses the diversity of generated 3D assets, it does not provide qualitative results demonstrating diversity for the same prompt. Including such examples would help assess the model’s ability to produce varied outputs under identical inputs.

Additional references:

[1] MeshLLM: Empowering Large Language Models to Progressively Understand and Generate 3D Mesh

[2] LLaMA-Mesh: Unifying 3D Mesh Generation with Language Models

**Questions:**

- The paper lacks detailed dataset statistics, particularly regarding the distribution between real-world and procedurally generated (PCG) data. Providing separate breakdowns, similar to those shown in Figure 9, for each data source would help understand the diversity and representativeness of the training data.

---

> ### Author Response · Authors · 2025-11-23
> **Response to Reviewer Kioq [Part 1/3]**
>
> Thank you for your thoughtful questions and suggestions. We have revised our paper accordingly. Here are our responses to your comments:
>
> > **Q1: Distribution statistics on two sets.**
>
> Thanks for this suggestion. We have added Fig. 13 presenting primitive type distributions for both our PCG dataset and real-world modular dataset. Additionally, we revised Fig. 14 (previously Fig. 9) to split statistical analysis into these two distinct subsets. We also corrected an error in the original Fig. 9's third sub-figure. These updated statistics are detailed in Appendix A.7.
>
> > **Q2: Similar in spirit to existing approaches and more discussion.**
>
> We appreciate you highlighting these relevant works. Indeed, our approach shares a conceptual similarity with [1,2] in leveraging autoregressive (AR) transformers for 3D generation. This common foundation underscores the growing interest in AR-based methods for structured 3D content creation. Below, we elaborate on these differences for clarity:
>
> - While MeshLLM [1] and LLaMA-Mesh [2] focus on adapting pre-trained LLM priors to understand and generate general 3D mesh representations (e.g., vertex-face meshes) in a progressive or unified manner, their emphasis is on broad generalization for basic mesh generation tasks. As a nascent area, their outputs, as shown in their results, often exhibit limited fidelity (e.g., constrained vertex/face counts), which may not suffice for high-complexity real-world applications. In contrast, our work prioritizes modular representations optimized for practical scenarios like user-generated content (UGC) and game development, even if it entails sacrificing some of the generalization afforded by LLM-inherited priors. Our modular data supports over one thousand primitives per asset, resulting in token sequences up to 5,000 tokens long—far exceeding the complexity typically handled in mesh-focused AR methods.
>
> - PrimitiveAnything [3], being a  primitive-based approach, similarly employs an AR transformer but for shape abstraction tasks. It operates on mesh inputs, using shape encoders and LLMs to decompose complex 3D shapes into primitive assemblies.  Our method, however, diverges by not relying on mesh-based inputs or abstractions; instead, we adopt a grid-based layout for modular primitives, a format prevalent in UGC game platforms (e.g., Minecraft, Roblox, and The Sims). This grid-based approach enables a stronger focus on the precise splicing and arrangement of internal structures, yielding benefits such as enhanced structural integrity, reduced computational overhead during assembly, and seamless integration with game engines for real-time editing and rendering.
>
> - Moreover, unlike [1,2], we do not depend on pre-trained LLM weights or priors. We directly tokenize modular primitives into a fixed vocabulary, allowing the AR transformer to learn fundamental building rules and compositional capabilities from scratch through sequence modeling. This design choice promotes interpretability and efficiency tailored to modular domains.
>
> **Contribution Clarification**: Our main contribution lies in proposing a modular 3D representation paired with custom-designed models that enable practical UGC development. As expanded in Appendix A.2, this representation surpasses traditional mesh formats by being efficient, lossless, directly engine-compatible, and accessible to non-expert users.
>
> We acknowledge that integrating LLM priors with modular representations remains a promising future direction. We have added the discussions on related works and representations in Section 2, Section 4.4.4, and Appendix A.2, and have made clarifications in the revised manuscript.
>
> *[1] MeshLLM: Empowering Large Language Models to Progressively Understand and Generate 3D Mesh. ICCV 2025.*
>
> *[2] LLaMA-Mesh: Unifying 3D Mesh Generation with Language Models. ArXiv 2024.*
>
> *[3] PrimitiveAnything: Human-Crafted 3D Primitive Assembly Generation with Auto-Regressive Transformer. SIGGRAPH 2025.*

---

> ### Author Response · Authors · 2025-11-23
> **Response to Reviewer Kioq [Part 2/3]**
>
> > **Q3: Whether the observed performance improvements come from the backbone LLM.**
>
> Thank you for this important question. As discussed in Q2, our model is **trained completely from scratch with random initialization**; we only borrow the LLaMA [4] decoder-only architecture for its proven scalability on long sequences, but do not load any pre-trained language modeling weights.
>
> To convincingly demonstrate this and respond to your suggestion for broader LLM-based benchmarking, we conducted the following additional experiment and summarized below:
>
> We explored fine-tuning pre-trained LLaMA-2 on our modular dataset with text-modality, but failed to produce satisfactory results, i.e., qualitative winrate below 5% over AssetFormer. We identify three key challenges:
> - **Complexity**: our buildings contain 1000+ primitives with intricate spatial constraints—far exceeding the simple meshes in [1,2];
> - **Implicit Representation**: our modular format encodes construction rules and spatial relationships implicitly, unlike explicit vertex-face meshes;
> - **Long Sequences**: token sequences reach up to 5,000, exceeding effective handling even with LongLoRA [5]. The exploratory experiment is updated in Appendix A.2.
>
> The experiment demonstrates our gains arise from the synergy between modular representation, tailored preprocessing, and custom model designs—not from pre-trained LLM priors. To rigorously validate the advantages of our modular representation, we conducted supplementary experiments (detailed in Appendix A.2 and Section 4.4.4) demonstrating its superiority over conventional 3D representations (mesh, VecSets, and sparse voxels). Specifically, we compared against mesh-based methods like MeshGPT [6] and native 3D representation frameworks. For further discussion on the ablation study involving native 3D representations, please refer to our response to Q4.
>
> *[4] LLaMA: Open and Efficient Foundation Language Models. ArXiv 2023.*
>
> *[5] LongLoRA: Efficient Fine-tuning of Long-Context Large Language Models. ICLR 2024.*
>
> *[6] MeshGPT: Generating Triangle Meshes with Decoder-Only Transformers. CVPR 2024.*

---

> ### Author Response · Authors · 2025-11-23
> **Response to Reviewer Kioq [Part 3/3]**
>
> > **Q4: Comparison with baseline text-to-3D models.**
>
> Thank you for this constructive suggestion. To strengthen the rigor of our comparisons, we have followed your advice and designed a targeted experiment to demonstrate that SOTA text-to-3D models struggle to generate or fit our modular data—primarily due to inherent limitations in their native data representations. This ablation study is now included in Section 4.4.4.
>
> - **Drawbacks of Existing Methods**: SOTA text-to-3D models [7,8,9,10] face inherent challenges in producing outputs with complex fine-grained details and well-preserved internal structures. While specialized training on such data may partially alleviate this issue, the core limitations are bounded by their reliance on native 3D representations (e.g., sparse voxels, VecSets). These models require watertight geometry preprocessing, where the "inside" and "outside" of 3D shapes must be clearly defined. However, this preprocessing step is often imperfect: it introduces unintended artifacts, distorts primitive geometries, or even alters the fundamental structure of modular components.
> - **Fitting Results**: To empirically validate that text-to-3D models cannot effectively fit our modular data, we conducted an overfitting experiment with Hunyuan3D 2.1. We use a subset of 10 modular building samples and fine-tune the model on this dataset. As shown in Fig. 5, critical issues emerged even before training: the watertight preprocessing step caused the loss of key primitives (e.g., ladders) and degraded the geometric details of remaining modules—both of which are essential for modular assets. Furthermore, the fine-tuning process proved infeasible for preserving modular integrity: the trained Hunyuan3D 2.1 produced outputs with severely compromised detail fidelity, as illustrated in Fig. 5(c) and (d). Based on our expertise in 3D generation particularly native geometry-based methods, we attribute these failures to two core limitations in existing literature: (1) the inflexibility of their native data representations for capturing modular structure, and (2) the unavoidable artifacts introduced by watertight preprocessing.
>
> *[7] Hunyuan3D 2.0: Scaling Diffusion Models for High Resolution Textured 3D Assets Generation. ArXiv 2025.*
>
> *[8] Structured 3D Latents for Scalable and Versatile 3D Generation. CVPR 2025.*
>
> *[9] SparseFlex: High-Resolution and Arbitrary-Topology 3D Shape Modeling. ICCV 2025.*
>
> *[10] Sparse Representation and Construction for High-Resolution 3D Shapes Modeling. ArXiv 2025.*
>
> > **Q5: Diversity for the same prompt.**
>
> We have supplemented a figure in Appendix A.5 showcasing multiple generated samples from the same text prompt, demonstrating the model’s diversity in modular building generation.

---

> ### Author Response · Authors · 2025-11-26
> **Follow-Up Rebuttal Discussion**
>
> Dear Reviewer Kioq,
>
> Thank you very much for your valuable feedback on our work. We appreciate the opportunity to engage in further discussion and hope that our response has addressed your concerns.
>
> In our rebuttal, we have clarified the key differences between our work and existing methods and have provided additional experimental results.
> If you find that we have satisfactorily addressed all your concerns, we would appreciate it if you would re-consider the score accordingly.
>
> Thank you again for your dedication to the review process.
>
>
> Best regards,
>
> Authors of Submission 18818

---

> > ### Comment · Reviewer_Kioq · 2025-11-26
> >
> > Thank you for your detailed response and for taking the time to address all my concerns. Your clarification has been very helpful, and I have updated my score accordingly.

---

> > > ### Author Response · Authors · 2025-11-27
> > > **Thanks for your comments!**
> > >
> > > We are delighted to hear that your concerns are addressed. Thanks for your constructive comments!

---

### Official Review · Reviewer_jBah · 2025-10-31

**Soundness:** 3
**Presentation:** 4
**Contribution:** 3
**Rating:** 8
**Confidence:** 3

**Summary:**

In this work, the authors introduces AssetFormer, which is a new type of autoregressive Transformer based framework. The authors design this for generating 3D assets in a modular way. Their approach focus on how assets are model from primitives and how they learn the distribution of these primitives for generative uses. The authors innovatively adapted token sequencing and decoding methods that is inspired by language models.

**Strengths:**

- Encouraging results.
- Theoretically well-written paper

**Weaknesses:**

- Can have more ablations.
- Can be good for the paper to have more expressive examples.

**Questions:**

What makes autoregressive modelling better than diffusion-based generation?

---

> ### Author Response · Authors · 2025-11-23
> **Response to Reviewer jBah**
>
> We sincerely thank the reviewer for the positive assessment. We would like to express our appreciation for your support of our paper and provide responses to all the raised questions:
> > **Q1: The advantages of autoregressive modelling over diffusion-based generation.**
>
> We consider it both from both native compatibility and practical feasibility perspectives:
> - Modular 3D data is inherently sequential, with modules encoding global context and location-based interactions. Hierarchical and spatial relationships emerge naturally during our DFS-based preprocessing, and these relations are not handcrafted but are intrinsic to modular sets. For example, assembling a LEGO model intuitively relies on neighboring part information to form meaningful structures: while no single "correct" assembly order exists, the sequence of part integration directly influences the final structure. This aligns seamlessly with the core mechanism of autoregressive modeling, which models sequence probabilities as an ordered chain of conditional distributions, making it inherently well-suited to capturing the logical flow and compositional structure of modular data. In contrast, general diffusion models struggle to encode such native sequential information in their latent spaces: modules lack flexible interactivity, and the iterative denoising process does not naturally align with the ordered, causal nature of modular assembly.
> - Practically, modular 3D representations lack the fixed-length structure that diffusion models typically target. While recent efforts [1] have adapted diffusion models for language modeling where data is naturally sequential and causal, these investigations remain nascent, and their applicability to more flexible modeling scenarios (e.g., our variable-length modular representations) remains unclear.
> - Despite the prevalence of VAEs and diffusion models in recent 3D geometry generation [2,3,4,5] (used to compress and learn VecSet- or sparse voxel-based representations), these approaches suffer from inferior topological routing fidelity, limited detail preservation, and weak controllability. This limitation is critical for UGC applications: results may deviate from intended themes, not just in visual quality, but also in geometric patterns where unwanted mesh complexity and routing artifacts require burdensome post-processing, reducing practical utility. In contrast, our autoregressive design is tailored to modular representations, enabling UGC-friendly generation with predefined primitives and intuitive editing capabilities.
>
> *[1] Large Language Diffusion Models. ArXiv 2025.*
>
> *[2] CLAY: A Controllable Large-scale Generative Model for Creating High-quality 3D Assets. TOG 2024.*
>
> *[3] Hunyuan3D 2.0: Scaling Diffusion Models for High Resolution Textured 3D Assets Generation. ArXiv 2025.*
>
> *[4] Structured 3D Latents for Scalable and Versatile 3D Generation. CVPR 2025.*
>
> *[5] SparseFlex: High-Resolution and Arbitrary-Topology 3D Shape Modeling. ICCV 2025.*
>
> > **Q2: Can have more expressive examples and ablations.**
>
> We sincerely appreciate the reviewer’s valuable feedback. In response, we have supplemented additional results and analyses in the revised manuscript, as detailed below:
> - **The Pre-trained Model for Emergent Editing**: We demonstrate zero-shot building editing without any task-specific fine-tuning. Leveraging pre-training on text-to-modular data, our model inherently learns the structural constraints and semantic relationships of modular buildings. As presented in Appendix A.4, we provide illustrative examples of this emergent editing capability: given a modular building representation, we first perform DFS-based token reordering as preprocessing, then remove unwanted primitives (e.g., existing roof structures) while the remaining tokens are the initial conditions. Notably, despite not being explicitly trained for editing nor exposed to the sometimes distinct token order patterns, the model successfully predicts the missing target primitives and completes the editing task.
> - **More Generated Results**: We have included additional modular building generations and integrated them into Unreal Engine (UE) for practical validation. As shown in Appendix A.6 (Fig. 11), the generated samples support intuitive texture mapping and seamless integration into UE workflows. This not only showcases the model’s editability but also demonstrates its applicability to real-world UGC and game development scenarios.
> - **Ablation Study on Modular Representation**: To validate the superiority of our modular representation, we conducted a comparative experiment with native diffusion models. Specifically, we converted our modular data into watertight meshes and used this processed data to overfit Hunyuan3D 2.1. The results (Fig. 5) show that the diffusion model fails to preserve the primitive structure of modular buildings and produces inferior details. This experiment and corresponding analysis are included in Section 4.4.4.

---

### Author Response · Authors · 2025-11-23
**General Response**

Dear ACs and Reviewers,

We sincerely thank all the reviewers for your insightful feedback and recognition of this work, especially for acknowledging the strengths of:

1. *Satisfactory results in modular 3D generation* (Reviewer jBah, Kioq),
2. *Our framework can be integrated in game engines and UGC, benefitting real-world applications* (Reviewer Kioq, mESg),
3. *The modular representation's advantages: efficiency, editability, interpretability, and seamless compatibility with autoregressive frameworks* (Reviewer Kioq, mESg),
4. *The dataset’s value for advancing modular 3D research and benefit to the community* (Reviewer Kioq, mESg, QQ43),
5. *Comparisons and ablation studies are sound and convincing* (Reviewer mESg, QQ43),
6. *The effectiveness of proposed modules (token ordering, sampling, and decoding)* (Reviewer Kioq, QQ43),
7. *Theoretically well-written* (Reviewer jBah).

---

Recent literature has seen a surge in autoregressive models for visual generation, alongside a shift toward representations that go beyond conventional native 3D formats—primarily built on VecSets and sparse voxels. Concurrently, the advancement of generative models for real-world applications has been increasingly encouraging and inspiring. Our work addresses these trends by designing a modular representation, tailored models, and specialized designs to support UGC and game development. We respectfully request reviewers to carefully evaluate our revisions and detailed individual responses.
In response to all reviewers’ insightful comments, we have refined the paper, supplemented experimental results, and expanded discussions in the revised version. All modifications are highlighted in blue for easy identification. Key updates include:

- We have included the experimental analysis comparing modular representation against native 3D representations in Section 4.4.4.
- We have added the experiments on emergent building editing in Appendix Section A.4.
- We have added the results showing the diversity on the same text prompts in Appendix Section A.5.
- We have added the more results in Appendix Section A.6.
- We have added the statistics of the primitives and modified the statistics of the text phrases in Appendix Section A.7.
- We have made several clarifications according to reviewers' suggestions.


Best regards,

Authors of Submission 18818

---

### Meta-Review · Area_Chair_LjpU · 2026-01-08

**Summary:**

The reviewers generally respond positively. The common concerns are usually related to experimental results. For example, Reviewer jBah request additional ablation. Reviewer Kioq request additional quantitative results, a better qualitative comparisons with a fine-tuning baseline, and more results to show diversity of the output. Reviewer mESg concerns about whether CLIP score is the right metrics. Reviewer mESg comments on the potential weakness in extensibility and generalization of the proposed method.

**Reviewer Concerns:**

The authors provides additional ablation studies and results in extending the paper to additional vocabulary and structure. Authors provide additional evidence to show the diversity of the output. They also clarify many questions regarding the positioning of the paper relative to other baselines and provided requested technical details.

In general I think authors' responses are well received and addressed most of reviewers concern.

**Reviewer Scores:**

Reviewer Kioq and mESg replied that they will increase the score and I believe Reviewer jBah will likely to maintain the positive rating.

---

### Decision · Program_Chairs · 2026-01-26

Accept (Poster)